# Effects of a non-standard information leaflet on patient recruitment in acute care: Embedded cluster-randomised controlled trial

Carolin Höckelmann[1]*, Marcelina Roos[1], Wiebke Müller[2], Martin N. Dichter[1], Sascha Köpke[1]

**1** University of Cologne, Faculty of Medicine and University Hospital Cologne, Institute of Nursing Science, Cologne, Germany, **2** University of Cologne, Faculty of Medicine and University Hospital Cologne, Institute of Medical Statistics and Computational Biology, Cologne, Germany

* carolin.hoeckelmann@uk-koeln.de

## Abstract

### Background

Recruiting study participants is a key component and major challenge in clinical research. Evidence shows that the design of written study information can influence recruitment success. However, there is a lack of research on the effectiveness and acceptability of different written study information.

### Objective

We aimed to investigate the effectiveness and acceptability of an information leaflet including an information video compared to a formal information letter in patient recruitment in acute care.

### Design

We conducted a cluster-randomised controlled trial embedded in a cross-sectional study ("Sleep Acute host study") addressing patients' sleep in hospitals.

### Methods

In a stratified random sample of hospitals, we allocated selected wards either to the intervention group (information leaflet including a link to an information video) or the control group (formal information letter) using external concealed randomisation. Adult patients hospitalised for at least 48 hours were eligible for participation. Our primary endpoint was recruitment success measured by the participation rate in the Sleep Acute host study. The secondary endpoint was acceptability of the written study information. Persons involved in the analyses were blinded concerning group allocation.

**Data availability statement:** The data are provided as part of the manuscript and its supporting information.

**Funding:** The author(s) received no specific funding for this work.

**Competing interests:** The authors have declared that no competing interests exist.

## Results

In total, 53 wards with 498 patients were cluster-randomised, 28 wards with 262 patients were allocated to the intervention group and 25 wards with 236 patients to the control group. The participation rate in the Sleep Acute host study was 51.1% (134 of 262) for the intervention and 47.5% (112 of 236) for the control group (OR 1.186 (0.698 to 2.013), p = 0.528). There were no significant differences concerning the acceptability of the written study information between both groups.

## Conclusions

More patients who received an information leaflet including a link to an information video participated in the host study, although the difference is not statistically significant.

## Study registration

DRKS-ID: DRKS00029707. Registered on 30. August 2022

## Introduction

Recruitment is defined as the identification of study participants, ensuring an adequate and/or representative sample [1]. Patient recruitment is a fundamental component of clinical studies but also a major challenge. In about half of the studies, it is estimated that the calculated sample size is not reached or not reached within the planned trial period or budget [2]. Failure to meet recruitment targets can result in underpowered studies, less valid study results, increased use of resources and premature study termination [1,3]. This impairs the value of study results and raises ethical questions [1,4].

Factors that influence study participation are multi-layered. In addition to the design and scope of a study, personal circumstances and potential benefits, informed consent [5], and informed decision-making are crucial for participants [6]. Heyrman et al. (2023) [7] showed a significant relationship between knowledge about the study and the willingness to participate. In addition to verbal information, it is required to provide written study information to adequately inform potential participants about the study's objectives, methodological approach, and potential risks of participation. However, formal information letters have repeatedly been identified as a potential barrier to participant recruitment, as comprehending scope and complexity is often described as too burdensome [8–10]. Pertinent recommendations to overcome these obstacles include, first, to provide written study information specifically designed for each target group [9], and second, to refrain from medical jargon [5]. Still, the evidence base for recruitment strategies is weak. A Cochrane review on strategies to improve recruitment listed 72 strategies, of which only three provided high-certainty evidence, of which two were not concerned with written study information and the third focused on user-tested written study information [11].

Based on the Medical Research Council START program, experts developed the Guideline for Reporting Embedded Recruitment Studies for interventions aiming to improve recruitment [12]. Now, various research groups used studies within trials (SWATs) to investigate the effects of different recruitment strategies in primary care. For example, Jolly et al. (2019) [13] investigated the effectiveness of providing access to a multimedia information resource alongside conventional printed study information. Madurasinghe et al. (2021) [14] conducted a meta-analysis of SWATs on optimised study information through user testing. These were compared with routine study information generated by the research teams. Results of six trials included in the meta-analysis indicated little to no difference in recruitment rates. Nevertheless, the data suggested that optimised study information leads to better comprehension.

Therefore, there is currently a lack of knowledge regarding the effectiveness of formal information letters compared to information leaflets for patient recruitment in acute care. Additionally, there is a particular lack of knowledge about the acceptability of written study information from users perspectives in order to derive needs and future recommendations for the design of written study information.

## Objectives

This study aims to investigate the effectiveness and acceptability of a study information leaflet including a link to an information video compared to a formal study information letter for patient recruitment in a sleep quality survey in acute care.

## Methods

### Study design

The host study "Sleep Acute" is a multicentre cross-sectional study aiming to describe hospital patients' subjective sleep quality, the prevalence of sleep disorders, and associated factors as well as strategies for sleep promotion. Patients received an 18-page paper-based questionnaire. We will publish the results of the host study elsewhere. This is a parallel group embedded cluster-randomised controlled trial (cRCT) to investigate the effectiveness and acceptability of an information leaflet including a link to an information video compared to a formal information letter. We chose the design of a cRCT to prevent contamination between groups.

The Ethics Committee of the German Society of Nursing Science approved the study (number: 20–029 a). The study registration was carried out retrospectively due to organisational reasons (DRKS-ID: DRKS00029707).

We followed the guidelines for reporting embedded recruitment trials, adapting the Consolidated Standards for Reporting Trials (CONSORT) guidelines for recruitment SWATs [12] and the guideline for cluster-randomised controlled trials (cRCT) [15] (S1 Table).

### Participants

The target population comprised patients aged ≥ 18 years who met the following inclusion criteria: (1) had stayed at least two nights in hospital, (2) were on a general ward, (3) were able to read German, and (4) were able to give informed consent. We excluded patients who were not present on the day of data collection during two visits by the researchers. For the purpose of the embedded cRCT, we included all potential participants invited to participate in the Sleep Acute host study employing the intention-to-treat (ITT) principle.

### Setting and recruitment

We conducted the study in acute care hospitals within the city of Cologne (Germany) and surrounding areas (50 km around Cologne). We included hospitals offering at least basic care and excluded military hospitals and preventive and rehabilitation hospitals. At ward level, we included general wards and excluded palliative care units, psychiatric wards, paediatric wards, emergency departments, functional areas, and intensive care units.

For the host study we identified hospitals based on hospital directories. For recruitment, we stratified hospitals according to their size (number of hospital beds) in four groups. We randomly selected hospitals using the randomisation software Random.org (https://www.random.org/sequences) and followed two successive recruitment strategies: First, we gradually recruited the hospitals on the list by contacting the nursing management. If hospitals declined to participate, we contacted the following hospital on the list. After a hospital had given its consent to participate, the nursing management selected the wards and randomisation was carried out. Depending on the hospital size, we planned the inclusion of two to eight wards per hospital (S5 Table).

Once we had selected the wards, we contacted the ward head of nursing by phone to give oral and written information about the study and to arrange an appointment for data collection.

We included all patients of the participating wards fulfilling our inclusion criteria and collected data on one day per ward. Nursing staff on the ward distributed written study information to eligible patients the day before or on the day of data collection and obtained permission for researchers to contact the patients.

### Intervention

The intervention consists of: (1) an information leaflet including a link to an information video for recruitment (IG) (S2 File); or (2) a formal information letter (CG) (S3 File).

Intervention group: We iteratively developed the paper-based information leaflet within the research team and in consultation with university's photo and graphics service (MedizinFotoKöln, https://medfak.uni-koeln.de/fakultaet/service/dienstleistungen-qualitaetssicherung/medizinfotokoeln), following current recommendations. Compared to the content of the formal information letter, we reduced the length and detail. Based on the recommendations by Antoniou et al. [8], we aimed to present the information leaflet in an appealing way, including a thematic image. In order to integrate an additional communication channel [16], the leaflet contained a QR code and a web link leading to an information video. In the video, the researchers introduced themselves and the research project in detail.

Control group: The design of the paper-based formal A4-size information letter was based on the specifications of the Ethics Committee of the German Society for Nursing Science (https://dg-pflegewissenschaft.de/). The first part provides general information on the study (background and objectives, procedure, benefits and risks), while the second part contains detailed information on data protection.

Both written study information used simple, easy-to-understand language [9] and were reviewed by the ethics committee in terms of their applicability.

### Outcomes and data collection

The primary endpoint was the participation rate, defined as the number of patients participating in the Sleep Acute host study divided by the number of patients invited to participate. We assessed the effectiveness comparing participation rates between the IG and CG. Our secondary endpoint that measures the acceptability of both recruitment approaches using a questionnaire is listed in Table 3.

To assess acceptability, we developed a questionnaire (S4 File) by completing the following steps: First, we conducted a comprehensive literature search to identify suitable questionnaires. As no adequate questionnaire existed, three nursing scientists with expertise in questionnaire development developed a questionnaire based on relevant literature in an iterative way. Subsequently, we conducted a pre-test with two people without a scientific background. Following the pre-test, we made minor adjustments to the wording of individual questions.

The final questionnaire consisted of three domains with 16 items: (1) evaluation of the written study information – eight closed questions (5-point rating scales); (2) evaluation of the written study information – three open-ended questions; and (3) sociodemographic characteristics (e. g., education and cultural background).

Together with a staff nurse on each ward, we extracted patients' sociodemographic characteristics (age and gender) and clinical data (specialty, admission diagnosis, operation(s), date and type of admission) from the patient files in order to minimise patient burden.

Patients received the written study information one day before data collection by a ward nurse. If ward nurses did not have enough time, the information was handed out on the day of data collection. The study team personally contacted all interested patients and provided information, answered questions, obtained patients' written informed consent, and handed out and collected paper-based questionnaires for both the host study and the embedded cRCT. Of patients who declined to participate in the study, only their anonymously recorded decision was used for data analysis. We collected completed questionnaires on the same or the following day. As the study required extensive staff resources, different researchers carried out recruitment and data collection. The number of researchers involved varied, depending on the number and size of wards.

We assigned code numbers to the questionnaires to identify the hospital, the ward, and temporarily the patient. We used a code list to coordinate the distribution and collection of questionnaires. At the end of each data collection, we anonymised the code list at patient level. Hospitals and wards remained pseudonymised. The code list was only accessible to the study team.

During the whole process, the nursing management and head of nursing were able to contact the research team in case of questions.

### Randomisation and blinding

We performed cluster-randomisation defining a cluster as a single participating ward within a hospital with all eligible patients admitted during data collection. Participating wards were randomly assigned to either intervention or control group on a 1:1 allocation for each hospital. To ensure allocation concealment, an external person, who was not involved in cluster assignment and in participant recruitment, performed randomisation using computer-generated randomisation lists. The external person informed the researchers of the cluster-randomisation by e-mail before data collection. Each ward had its own code number for identification. It was not possible to blind the researchers administering the written study information to group allocation, but researchers involved in the data analyses were blinded to the cluster assignment. Participants were unaware of taking part in an embedded cRCT.

### Sample size

Following methodological recommendations, we conducted no formal sample size calculation for the embedded cRCT, given that the sample size is constrained by the number of participants approached in the Sleep Acute host study [17].

Due to the study's exploratory nature, we also did not perform a formal sample size calculation for the Sleep Acute host study. However, we were guided by a similar survey [18] and assumed a sample size of 1,040 patients from 104 wards in 31 hospitals (i.e., 10 questionnaires/ward).

### Statistical analysis

We analysed the data both quantitatively and qualitatively.

**Quantitative data analysis.** We derived the primary analysis population from the intention-to-treat principle (no exclusions). To analyse the primary endpoint 'participation', we used random effects logistic regression adjusted for cluster using the Stata command 'xtlogit, vce (robust)' with cluster as panel variable and group as covariate (alpha = 0.05). We conducted bivariate linear regressions with the secondary endpoint as dependent variable and group as independent variable (standard error adjusted for clusters) to compare the acceptability of the recruitment

approaches between groups. Secondary, we applied all analyses on the per protocol set. The per protocol set included all patients who received the allocated intervention or control, respectively. We described continuous variables with mean (standard deviation (SD)) and median (first quartile, third quartile), categorical variables with absolute and relative frequencies. We conducted quantitative analyses using SPSS Version 28 [19] and Stata Version 18.0 [20].

**Qualitative data analysis.** We analysed open-ended questions in a structured process based on Mayring [21] using MAXQDA version 22.0.0 [22]. One person categorised the data and a second person checked the analysis for comprehensibility and completeness.

## Results

### Sample

Recruitment and data collection took place from 6 May 2021 to 29 June 2022. Out of 60 randomly selected hospitals, we initially informed the nursing management about the research project by e-mail and phone. 13 hospitals and 53 wards agreed to participate in the study (Fig. 1). Of a mean 25 patients per ward, we recruited on average five patients per ward, resulting in 498 participants of whom 262 were allocated to the IG and 236 to the CG. Fig 1 outlines the participant flow.

Patient characteristics were balanced between groups for both analysed and participating patients, although there were slightly more women in the IG than in the CG (Table 1).

### Participation rate (Primary Outcome)

In the IG 134 of 262 patients participated in the Sleep Acute host study, a participation rate of 51.1%. In the CG 112 of 236 patients participated, a participation rate of 47.5% (Table 2a), i.e. patients in the IG had a non-significant higher chance of participating than patients in the CG (cluster-adjusted OR=1.186, 95% CI 0.698 to 2.013, p = 0.528). The intra-class correlation coefficient (ICC) was 0.098 (95% CI 0.032 to 0.266).

The per-protocol analysis was similar to the intention-to-treat analysis, also showing a non-significant effect of the information leaflet on participation rates (cluster-adjusted OR=1.475, 95% CI, 0.716 to 3.038, p = 0.292). The intra-class correlation coefficient (ICC) was 0.108 (95% CI, 0.027 to 0.348) (Table 2b).

### Acceptability

Two-hundred of 498 (response rate 43.2%) patients completed the questionnaire to evaluate the written study information (IG = 111; CG = 89).

With regard to the acceptability of the written study information, there were no significant differences between groups. Participants in the IG rated the information leaflet including an information video not better than participants in the CG, who rated the formal information letter. In general, participants rated both written study information rather positive (Table 3). During data collection, the information video was accessed through the QR code 15 times.

The answers of the open-ended questions also demonstrated positive ratings with suggestions for change in both groups. Participants noted, for example, a low information content regarding study-related details with a simultaneous desire for more concise and shorter information. They also criticised the partial lack of simple language. However, it is noticeable that the participants appreciated the information leaflet for its modern, compact design focusing on essential information as well as the QR code, leading to the information video. Some participants criticised the leaflet design, particularly the selection of the photo on the front page.

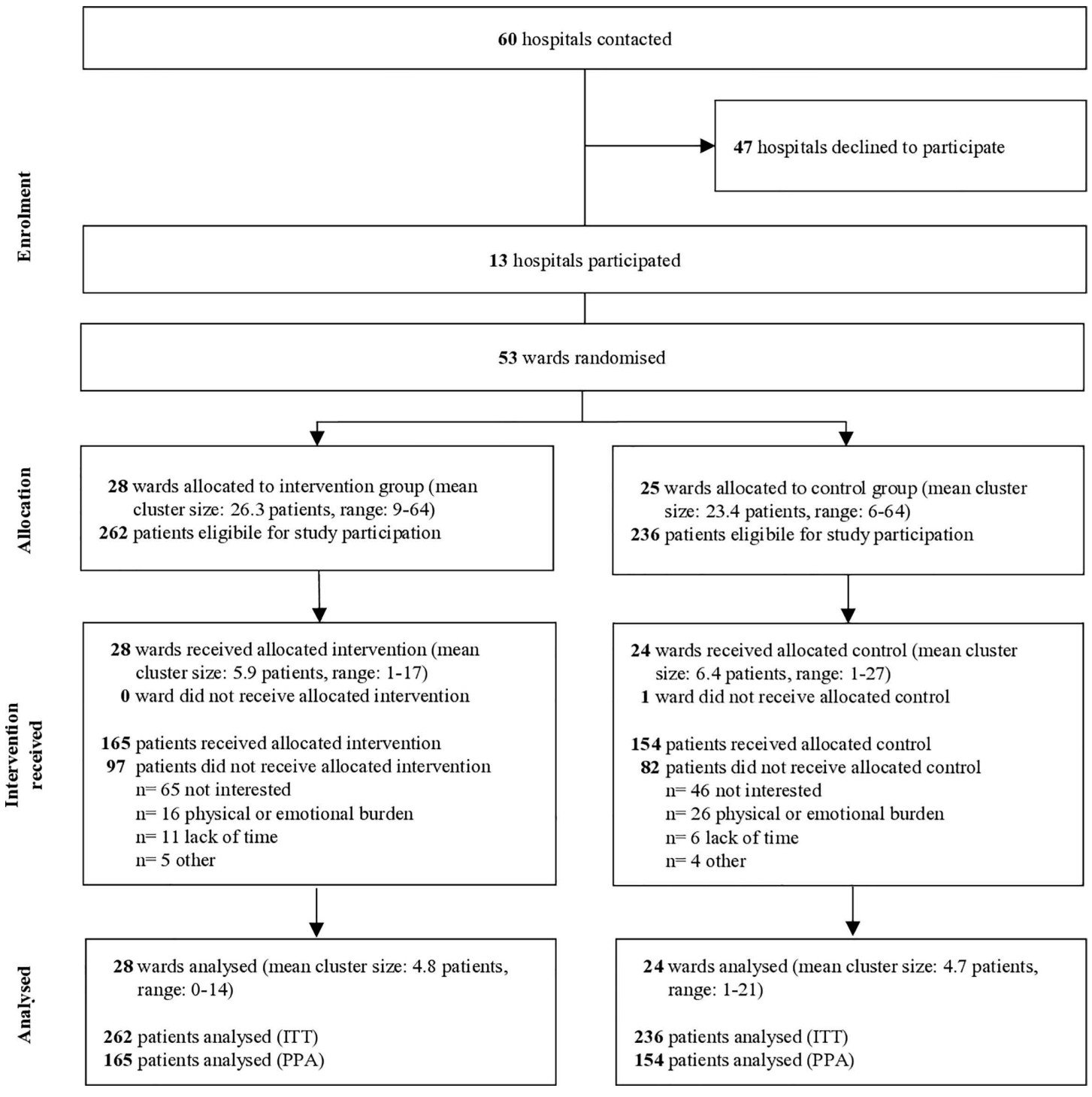

**Fig 1. Flowchart.** (ITT) Intention-to-Treat Analysis. (PPA) Per-Protocol Analysis.

**Table 1. Characteristics of analysed (n = 498) and participating patients (n = 246).**

| | Analysed patients in randomised wards | | Participating patients in randomised wards | |
|---|---|---|---|---|
| | Intervention<br>n = 262 | Control<br>n = 236 | Intervention<br>n = 134 | Control<br>n = 112 |
| **Sex, n (%)*** | | | | |
| Female | / | / | 81 (60.4) | 48 (42.9) |
| Male | / | / | 48 (35.8) | 58 (51.8) |
| Missing data | / | / | 5 (3.7) | 6 (5.4) |
| **Age*** | | | | |
| Mean years (SD) | / | / | 60.26 (±17.45) | 59.01 (±16.85) |
| Median (IQR) | / | / | 62 (49.75;73) | 61.5 (47; 71.75) |
| Missing data (n (%)) | / | / | 12 (9) | 16 (14.3) |
| **Length of stay** | | | | |
| Mean days (SD) | 10.6 (± 11.9) | 8.6 (± 9.45) | 10.1 (±10.66) | 8.04 (±.7.74) |
| Median (IQR) | 6 (4; 12) | 6 (4; 9.75) | 6 (3; 11) | 6 (4; 9) |
| Missing data (n (%)) | 7 (2.7) | 12 (5.1) | / | / |
| **Admission type, n (%)** | | | | |
| Elective/planned | 121 (46.2) | 112 (47.5) | 67 (50) | 62 (55.4) |
| Emergency | 125 (47.7) | 106 (44.9) | 62 (46.3) | 49 (43.8) |
| Unclear/unknown | 8 (3.1) | 6 (2.5) | 5 (3.7) | 1 (0.9) |
| Missing data | 8 (3.1) | 12 (5.1) | / | / |
| **Speciality, n (%)** | | | | |
| Internal medicinea | 95 (36.3) | 73 (30.9) | 35 (26.1) | 46 (41.1) |
| Surgeryb | 114 (43.5) | 128 (54.2) | 72 (53.7) | 54 (48.2) |
| Otherc | 31 (11.8) | 21 (8.9) | 21 (15.7) | 11 (9.8) |
| Missing data | 22 (8.4) | 14 (5.9) | 6 (4.5) | 1 (0.9) |
| **Previous surgery, n (%)** | | | | |
| Yes | 115 (43.9) | 112 (47.5) | 73 (54.5) | 52 (46.4) |
| No | 140 (53.4) | 112 (47.5) | 61 (45.5) | 60 (53.6) |
| Missing data | 7 (2.7) | 12 (5.1) | / | / |

*Data only available for participants in the host study

**Table 2. a. Participation rate (Sleep Acute host study, n = 498) – Intention-to-Treat analysis. b. Participation rate (Sleep Acute host study, n = 319) – Per-Protocol Analysis.**

| | Total<br>(n = 498) | Intervention<br>(n = 262) | Control<br>(n = 236) |
|---|---|---|---|
| Participant, n (%) | 246 (49.4) | 134 (**51.1**) | 112 (**47.5**) |
| Non-Participant, n (%) | 252 (50.6) | 128 (48.9) | 124 (52.5) |
| ICC (rho), p-value[a] | | 0.098 [−0.082 to 0.161] | |
| Odds Ratio (IG vs. CG) [95% CI], p-value[a] | | 1.186 [0.698 to 2.013], p = 0.528 | |
| | Total<br>(n = 319) | Intervention<br>(n = 165) | Control<br>(n = 154) |
| Participant, n (%) | 246 (77.1) | 134 (81.2) | 112 (72.7) |
| Non-Participant, n (%) | 73 (22.9) | 31 (18.8) | 42 (27.3) |
| ICC (rho), p-value[a] | | 0.108 [0.027 to 0.348] | |
| Odds Ratio (IG vs. CG) [95% CI], p-value[a] | | 1.475 [0.716 to 3.038], p = 0.292 | |

CG: Control Group; ICC: Intra-class Correlation Coefficient; IG: Intervention Group

[a]Random effects logistic regression with cluster as panel variable and group as covariate (alpha = 0.05)

**Table 3. Acceptability (n = 200).**

| | Intervention n=111 | Control n=89 | β (CI 95%), p-value[a] |
|---|---|---|---|
| 1. How do you rate the receipt of written information about the study? | | | |
| Mean (SD) | 4.22 (± 0.95) | 4.22 (± 0.762) | -0.003 (-0.31 to 0.31), 0.982 |
| Median (IQR) | 4.5 (4; 5) | 4 (4; 5) | |
| n.a. (n(%)) | - | 2 (2.2) | |
| 2. Did the written study information encourage you to participate in the study?[c] | | | |
| Mean (SD) | 3.12 (± 1.67) | 3.36 (±1.3) | -0.02 (-0.8 to 0.32), 0.393 |
| Median (IQR) | 3.75 (2; 4.5) | 4 (3; 4) | |
| n.a. (n(%)) | 1 (0.9) | 2 (2.2) | |
| 3. How do you rate the written study information overall?[b] | | | |
| Mean (SD) | 3.97 (± 0.96) | 3.92 (± 0.92) | 0.05 (-0.36 to 0.47), 0.792 |
| Median (IQR) | 4 (4; 5) | 4 (3.875; 4.5) | |
| n.a. (n(%)) | - | 3 (3.4) | |
| 4. How do you rate the format and design of the written study information?[b] | | | |
| Mean (SD) | 3.77 (± 0.96) | 3.78 (± 0.8) | -0.11 (-0.19 to 0.4), |
| Median (IQR) | 4 (3; 4.625) | 4 (3; 4) | |
| n.a. (n(%)) | 1 (0.9) | 4 (4.5) | |
| 5. How do you rate the written study information in terms of its comprehensibility?[d] | | | |
| Mean | 4.11 (± 0.9) | 4 (± 0.95) | 0.12 (-0.19 to 0.4),0.465 |
| Median (IQR) | 4 (4; 5) | 4 (3; 5) | |
| n.a. (n(%)) | 4 (3.6) | 4 (4.5) | |
| 6. How do you rate the completeness of the written study information?[e] | | | |
| Mean (SD) | 4.01 (± 0.87) | 4.02 (± 0.83) | -0.01 (-0.26 to 0.25), 0.937 |
| Median (IQR) | 4 (3.5; 5) | 4 (3; 5) | |
| n.a. (n(%)) | 4 (3.6) | 6 (6.7) | |
| 7. How do you rate the comprehensiveness of the written study information?[f] | | | |
| Mean (SD) | 4.07 (± 0.9) | 3.91 (± 0.94) | 0.17 (-0.07 to 0.41), 0.163 |
| Median (IQR) | 4 (3.5; 5) | 4 (3; 5) | |
| n.a. (n(%)) | 2 (1.8) | 5 (5.6) | |
| 8. How do you rate the content of the written study information in terms of its relevance?[g] | | | |
| Mean (SD) | 3.86 (± 0.99) | 3.92 (± 0.88) | -0.07 (-0.36 to 0.23), 0.644 |
| Median (IQR) | 4 (3; 5) | 4 (3.125; 5) | |
| n.a. (n(%)) | 4 (3.6) | 5 (5.6) | |

[a] bivariate linear regression (standard error adjusted for clusters) with group as independent variable and each secondary endpoint (1.-8.) as dependent variable

Questions were answered using a 6-point scale, scored as follows:

[b] 0 = very negative to 5 = very positive;

[c] 0 = not at all to 5 = greatly;

[d] 0 = not at all understandable to 5 = very understandable;

[e] 0 = not complete to 5 = complete;

[f] 0 = not appropriate at all to 5 = perfectly adequate;

[g] 0 = not relevant at all to 5 = very relevant

## Discussion

Comprehensible written study information is essential to ensure informed choice for participation in a study. Although written study information for recruitment is subject to ethics committee requirements, researchers have an important role in the development and design, including the appropriate use of plain language [23].

In this embedded cRCT, we tested the effectiveness and acceptability of a non-standard information leaflet including an information video compared to a formal information letter on patient recruitment in a sleep quality study in acute care. We found no significant difference in participation rates between the two written study information, although the IG participation rate was higher with absolute differences of 3.6% (ITT) and 8.5% (PPA). Thus, the use of an information leaflet does appear to improve participant recruitment. In terms of acceptability, we found some indications that study participants perceive certain elements of an information leaflet positively.

Our results complement existing research. A Cochrane review on strategies to improve recruitment indicates no statistical significance impact on recruitment by a brief study information letter compared to a usual study information letter (moderate certainty evidence). Providing information by video versus standard approaches alone made little or no difference to recruitment either (very low certainty evidence) [11]. A recent meta-analysis of SWATs supports this finding, indicating that the use of multimedia alongside written study information results in little or no difference in recruitment rates [24].

One reason for the limited effectiveness and acceptability of the information leaflet including an information video might be the high average participant age of around 60 years, which is associated with a lower ability of digital technologies usage [25]. Jolly et al. (2019) [13] discuss the extent to which the target group influences the success of multimedia information sources. The participants in their host trial were predominantly retired males over 70 years for whom additional access to a multimedia information source did not result in any advantages in recruitment [13]. In addition, it should be noted that participants evaluated the written study information without any comparison, suggesting that patients who received the formal information letter were unaware that alternative formats may be available. This might explain the not significant results of acceptability between the intervention and control group.

People who are hospitalised due to acute illness are a hard-to-reach target group. The lack of interest in study participation of hospital staff is another significant factor influencing recruitment [26]. Our study underlines the challenges of recruiting participating wards and patients. However, the nature of the host study also has to be taken into account when interpreting the results of this embedded cRCT. The research project did not receive any funding, therefore recruitment had to be carried out without study nurses or incentives for hospital staff or patients. Further, in addition to the written study information, participants received an 18-page questionnaire to assess sleep quality and a 4-page questionnaire to assess acceptability of the study information. The amount of study materials might have led to a high burden for study participants, influencing the willingness to participate [23]. It remains unclear to what extent the involvement of study nurses, the use of incentives and a decreased amount of study materials would have affected recruitment.

### Strengths and limitations

One of the main strengths of our study is that, apart from results for participation rates, we can also draw conclusions about the acceptability of both study information. This provides in-depth insights into the perceptions of the target population, which can be used for the design of future written study information and further research on recruitment. Another strength of our study is that we selected hospitals and wards in and around Cologne randomly. Thus, we have included both urban and rural regions and a variety of different specialties. We assume that our results are representative for the acute care context in Germany. Another strength of our study is the inclusion of an informational video as part of the leaflet, accessible via QR code. This video offers an easy way to access study information without the need to read,

potentially increasing accessibility and participation. However, the video was rarely accessed and we cannot rule out multiple access by the same person or access by persons outside the target group.

A limitation of our study is the lack of psychometric evaluation of the questionnaire. Although we developed the instrument through an extensive process, we refrained from conducting a formal validation for pragmatic reasons. This should be addressed in any future use of the questionnaire. In addition, another limitation is the lack of involvement of the target group in the development of the information leaflet. As a result, we had no information about the strengths and weaknesses from the target group's perspective and could not ensure that the information leaflet fits its purpose [27]. However, the Cochrane review suggests a small impact of user-tested study information on recruitment (high certainty evidence) [11]. In addition, the design of the leaflet follows current expert recommendations and the multi-round development process involved design experts and researchers with extensive experience in the creation of study information and in clinical patient care in hospitals.

Although we randomly selected hospitals and wards in and around Cologne, we were only able to recruit a quarter of the assumed sample size, which was to some extent related to the Covid-19 pandemic. We had initially planned to recruit 1040 patients from 104 wards in 31 hospitals. Due to recruitment problems, we reduced the planned sample size by half. Recruitment lasted 14 months and ended prematurely after the inclusion of 50% of the modified sample size (246 patients, 52 wards and 13 hospitals), which leads to a reduced power and therefore might have an impact on the validity of the results of the embedded cRCT. The promising results suggest that a larger number of participants would have led to significant results. Our own challenging experiences in recruitment emphasize the relevance and necessity of scientific research approaches to improve patient recruitment in acute care.

Due to the extensive staff resources required to conduct the study, different researchers carried out recruitment and made personal contact with eligible patients during the handover of study information. Subsequently, different procedures and individual effects with regard to recruitment success cannot be completely ruled out as the success of patient recruitment is linked, for example, to the education and training of researchers [9,26,28]. This procedure is associated with a risk of performance bias. However, we developed a protocol for conducting the data collection and trained all data collectors in advance to achieve a standardised procedure.

In addition, the lack of flexibility of the research team has an impact on participation rates [26]. We recruited during the morning shift. At this time, patients were often involved in nursing and medical procedures, as examinations and therapies. This indicates the importance of flexibility considering hospital structures and ward workflows.

If ward nurses were unable to distribute written study information on the day before data collection due to time constraints, researcher provided it to the patients on the day of data collection. As a result, patients had varying amounts of time to review the study materials, which may have influenced their decision to participate [5]. Unfortunately, we are unable to provide valid data on the date of distribution of study information.

Due to organizational reasons within the study team, the study was registered retrospectively. However, the study protocol had been approved by the ethics board before the study commenced. Also, as this is a cross-sectional trial with a relatively simple study design, we do not consider this a major problem. Another limitation of our study is the lack of age and gender data for all participants, as these were only collected in the host study and are missing for those in Sleep-Recruit only or for eligible individuals who did not receive the intervention.

## Conclusion

In conclusion, this study adds to the evidence of studies aiming to assess the impact of different recruitment interventions to improve the process of study recruitment. Our findings suggest a potential impact of an information leaflet including a link to an information video on patient recruitment in acute care. However, larger studies are needed to confirm these findings.

Improving evidence on recruitment has the potential to increase participation rates, improve the internal and external validity of study results and increase the proportion of studies that are completed on time. Based on the assumption that people react differently to recruitment approaches, future research should focus on the socio-demographic diversity. This information is particularly important in order to represent a wide range of research participants in a sample and to be able to specifically involve certain groups of people in research [29]. For this purpose, embedded studies are particularly suitable. SWATs could rapidly expand the evidence base in recruitment strategies and thus reduce the evidence gap [30].

## Supporting information

**S1 Table. CONSORT Checklists. Consolidated Standards for Reporting Trials (CONSORT) for recruitment SWATs and cRCT.**
(DOCX)

**S2 File. Formal study information letter.**
(PDF)

**S3 File. Study information leaflet.**
(PDF)

**S4 File. Questionnaire acceptability.**
(PDF)

**S5 Table. Planned sample size (stratification by hospital size) and planned sample size after adjustment (stratification by hospital size).**
(DOCX)

**S6 File. SPSS Row data.**
(SAV)

**S7 File. Study protocol German/English.**
(PDF)

## Acknowledgments

We thank all patients who participated in this study and all team members for their support in recruitment, especially Verena von der Lühe, Richard Dano, Mareike Löbberding and Petra Schirk. We also thank Eva Diegel and Beyza Sadat for their support in data entry.

## Author contributions

**Conceptualization:** Marcelina Roos, Martin N. Dichter, Sascha Köpke.

**Data curation:** Carolin Höckelmann, Marcelina Roos.

**Formal analysis:** Wiebke Müller.

**Investigation:** Carolin Höckelmann, Marcelina Roos.

**Methodology:** Marcelina Roos, Martin N. Dichter, Sascha Köpke.

**Project administration:** Martin N. Dichter, Sascha Köpke.

**Supervision:** Martin N. Dichter, Sascha Köpke.

**Validation:** Carolin Höckelmann, Marcelina Roos, Wiebke Müller, Martin N. Dichter, Sascha Köpke.

**Visualization:** Carolin Höckelmann, Marcelina Roos.

**Writing – original draft:** Carolin Höckelmann, Marcelina Roos.

**Writing – review & editing:** Carolin Höckelmann, Marcelina Roos, Wiebke Müller, Martin N. Dichter, Sascha Köpke.

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
