## [Decision Letter · Decision Letter 0]

Dear Dr. Höckelmann,

Thank you for submitting your manuscript to PLOS ONE. After careful consideration, we feel that it has merit but does not fully meet PLOS ONE’s publication criteria as it currently stands. Therefore, we invite you to submit a revised version of the manuscript that addresses the points raised during the review process.

We look forward to receiving your revised manuscript.

Kind regards,

Maher Abdelraheim Titi

Academic Editor

PLOS ONE

Journal Requirements:

2. We note that you have selected “Clinical Trial” as your article type. PLOS ONE requires that all clinical trials are registered in an appropriate registry (the WHO list of approved registries is at https://www.who.int/clinical-trials-registry-platform/network/primary-registries"
https://www.who.int/clinical-trials-registry-platform/network/primary-registries and more information on trial registration is at http://www.icmje.org/about-icmje/faqs/clinical-trials-registration/). Please state the name of the registry and the registration number (e.g. ISRCTN or ClinicalTrials.gov) in the submission data and on the title page of your manuscript. a) Please provide the complete date range for participant recruitment and follow-up in the methods section of your manuscript. b) If you have not yet registered your trial in an appropriate registry, we now require you to do so and will need confirmation of the trial registry number before we can pass your paper to the next stage of review. Please include in the Methods section of your paper your reasons for not registering this study before enrolment of participants started. Please confirm that all related trials are registered by stating: “The authors confirm that all ongoing and related trials for this drug/intervention are registered”. Please see http://journals.plos.org/plosone/s/submission-guidelines#loc-clinical-trials for our policies on clinical trials.

3. In the online submission form, you indicated that [The data underlying the results presented in the study are available from the authors.].

Reviewers' comments:

Reviewer's Responses to Questions

**Comments to the Author**

1. Is the manuscript technically sound, and do the data support the conclusions?

Reviewer #1: Partly

Reviewer #2: Yes

2. Has the statistical analysis been performed appropriately and rigorously?

Reviewer #1: Yes

Reviewer #2: Yes

3. Have the authors made all data underlying the findings in their manuscript fully available?

Reviewer #1: Yes

Reviewer #2: Yes

4. Is the manuscript presented in an intelligible fashion and written in standard English?

Reviewer #1: Yes

Reviewer #2: Yes

Reviewer #1: The purpose of this research was to investigate the effectiveness and acceptability of an information leaflet including an information video compared to a formal information letter in patient recruitment in acute care.

Accordingly the investigators conducted a cluster-randomised controlled trial embedded in a cross sectional study (“Sleep Acute host study”) addressing patients’ sleep in hospitals. They allocated selected wards either to the intervention group (information leaflet including a link to an information video) or the control group (formal information letter) using external concealed randomization. The design was well thought out and the methodology was well described.

The primary endpoint was recruitment success measured by the participation rate in the Sleep Acute host study. The secondary endpoint was acceptability of the written study information. Persons involved in the analyses were blinded concerning group allocation.

Due to the study’s exploratory nature, there was no formal sample size calculation for the embedded cRCT, given that the sample size is constrained by the number of participants approached in the Sleep Acute host study, although a more rigorous hypothesized effect size would have been ideal . Also given the exploratory and mostly descriptive nature of the presentation it appears that the number per group (analyzed and participated) is adequate for the descriptive results reported.

Analysis was quantitative and qualitative. To analyze the primary endpoint ‘participation’, they used random effects logistic regression adjusted for cluster. Exactly what was the random effect or was it cluster? The odds ratios indicated no difference between the two groups. They also conducted bivariate linear regressions with the secondary endpoint . The explanation of the approaches was reasonable for this type of data. Qualitative data analysis involved the usual open-ended questions in a structured process. Comparison across questions is noted in Table 3. Differences of acceptability was not evident examining the confidence intervals of the slopes.

They do conclude that the use of an information leaflet does appear to improve participant recruitment. In terms of acceptability, they found some indications that study participants perceive certain elements of an information leaflet positively. Such was the mainly descriptive nature of the cautionary presentation.

Reviewer #2: Thank you for the opportunity to comment on this interesting, nested study (it is not a formal SWAT as the host study is not a trial). It is clearly described and the methods look rigorous, and is based on a large sample of wards and patients (although the sample was smaller than expected). I only had a few comments. The study is interesting as it is in a hospital context.

The first line of the introduction seems to define ‘recruitment’ to include ‘retention’, whereas these are generally distinguished in the literature.

Can they clarify the consent procedures for the nested study? It is mentioned 3 times (page 4, 7 and 8). I think they just need to state in one place that people consented to the sleep survey, but not the embedded study.

On page 7, they suggest that extensive development is a basis for validity. I am not sure that is the consensus view and I do not think they need to make that claim. They just need to be clear what they did do for validity assessment, and what they did not do – and let readers judge the validity.

Can they clarify the proportion of patients who received the information one day before, and those who got it on the day (page 6)? I think that would be useful context.

I think it would be useful to comment on the baseline level of participation (which I think is 20% of eligible patients being approached at all, and around 50% of those approached taking part). Am I right in thinking that the ‘window’ for people to take part once asked was 24 hours (‘same or following day’)? That might have limited the potential impact of the intervention.

Do they know how many people visited the website through the QR code? I think that would be useful information to provide. If they do not have that (and I have been in that position), I would be explicit.

I think it would help if the intervention was better labelled. The title says ‘information leaflet’ and does not mention the full intervention. I think many readers may assume that a ‘leaflet’ is pretty standard and it would be helpful to label the intervention in such a way that highlights its differences from ‘usual care’ (which I assume is length/detail, and the QR link).

Apologies for the self-citation but Madurasinghe et al. BMC Med. 2023 Nov 8;21(1):425. doi: 10.1186/s12916-023-03081-5 might be a relevant comparison as these studies involved links to multimedia.

Can they clarify the reason for the missing data in Table 1? Is it right that they did not have age and gender data for all patients?

There is a statement in the discussion (page 15) which talk about ‘potential impact’ and the abstract says that the intervention group showed larger effects but that it was ‘not significant’. Formally this is true, but it is a particular framing – ‘no statistically significant effect of the intervention’ would be equally valid. I would go for the most neutral framing possible I think

**Do you want your identity to be public for this peer review?** For information about this choice, including consent withdrawal, please see our Privacy Policy

Reviewer #1: No

Reviewer #2: **Yes: ** Peter Bower

---

## [Author Response · Author response to Decision Letter 1]

21 May 2025

Thank you for the review. You can find my response in the attached Word document 'Response to Reviewer'.

---

## [Decision Letter · Decision Letter 1]

Effects of a non-standard information leaflet on patient recruitment in acute care: Embedded cluster-randomised controlled trial

PONE-D-25-02678R1

Dear Dr. Höckelmann,

We’re pleased to inform you that your manuscript has been judged scientifically suitable for publication and will be formally accepted for publication once it meets all outstanding technical requirements.

Kind regards,

Maher Abdelraheim Titi

Academic Editor

PLOS ONE

Additional Editor Comments (optional):

The authors successfully addressed all comments from both reviewers, demonstrating that they were able to resolve all concerns raised during the peer review process.

Reviewers' comments:

Reviewer's Responses to Questions

**Comments to the Author**

Reviewer #1: All comments have been addressed

2. Is the manuscript technically sound, and do the data support the conclusions?

Reviewer #1: (No Response)

3. Has the statistical analysis been performed appropriately and rigorously?

Reviewer #1: (No Response)

4. Have the authors made all data underlying the findings in their manuscript fully available?

Reviewer #1: (No Response)

5. Is the manuscript presented in an intelligible fashion and written in standard English?

Reviewer #1: (No Response)

Reviewer #1: (No Response)

**Do you want your identity to be public for this peer review?** For information about this choice, including consent withdrawal, please see our Privacy Policy

Reviewer #1: No

---

## [Editor Report · Acceptance letter]

PONE-D-25-02678R1

PLOS ONE

Dear Dr. Höckelmann,

I'm pleased to inform you that your manuscript has been deemed suitable for publication in PLOS ONE. Congratulations! Your manuscript is now being handed over to our production team.

Kind regards,

on behalf of

Dr. Maher Abdelraheim Titi

Academic Editor

PLOS ONE